# The Antiviral Effect of *Isatis* Root Polysaccharide against NADC30-like PRRSV by Transcriptome and Proteome Analysis

**DOI:** 10.3390/ijms23073688

**Published:** 2022-03-28

**Authors:** Dike Jiang, Ling Zhang, Guangheng Zhu, Pengfei Zhang, Xulong Wu, Xueping Yao, Yan Luo, Zexiao Yang, Meishen Ren, Xinping Wang, Sheng Chen, Yin Wang

**Affiliations:** 1Key Laboratory of Animal Diseases and Human Health of Sichuan Province, College of Veterinary Medicine, Sichuan Agricultural University, Chengdu 611130, China; 2020103005@stu.sicau.edu.cn (D.J.); 2018303051@stu.sicau.edu.cn (G.Z.); 2018103006@stu.sicau.edu.cn (P.Z.); 13577@sicau.edu.cn (X.Y.); 41187@sicau.edu.cn (Y.L.); 13643@sicau.edu.cn (Z.Y.); 14773@sicau.edu.cn (M.R.); 2College of Veterinary Medicine, Jilin University, Changchun 130012, China; zhangl15@mails.jlu.edu.cn; 3Branch of Animal Husbandry and Veterinary Medicine, Chengdu Agricultural College, Chengdu 611130, China; yaanwuxl@163.com; 4Department of Infectious Diseases and Public Health, Jockey Club College of Veterinary Medicine and Life Sciences, City University of Hong Kong, Hong Kong 999077, China; shechen@cityu.edu.hk

**Keywords:** *isatis* root polysaccharide, NADC30-like porcine reproductive and respiratory syndrome virus, transcriptomic and proteomic analysis

## Abstract

(1) Background: In recent years, the porcine reproductive and respiratory syndrome virus (PRRSV) has become a virulent pathogen that has caused devastating diseases and economic losses worldwide in the swine industry. *IRPS* has attracted extensive attention in the field of virology. However, it is not clear that *IRPS* has an antiviral effect on PRRSV at gene and protein levels. (2) Methods: We used transcriptomic and proteomic analysis to investigate the antiviral effect of *IRPS* against PRRSV. Additionally, a microbiome was used to explore the effects of *IRPS* on gut microbes. (3) Results: *IRPS* significantly extenuated the pulmonary pathological lesions and inflammatory response. We used transcriptomic and proteomic analysis to investigate the antiviral effect of *IRPS* against PRRSV. In the porcine model, 1669 differentially expressed genes (DEGs) and 370 differentially expressed proteins (DEPs) were identified. Analysis of the DEG/DEP-related pathways indicated immune-system and infectious-disease (viral) pathways, such as the NOD-like receptor (NLR) signaling pathway, toll-like receptor (TLR) signaling pathway, and Influenza A-associated signaling pathways. It is noteworthy that *IRPS* can inhibit NLR-dependent gene expression, then reduce the inflammatory damage. *IRPS* could exert beneficial effects on the host by regulating the structure of intestinal flora. (4) Conclusions: The antiviral effect of *IRPS* on PRRSV can be directly achieved by omics techniques. Specifically, the antiviral mechanism of IPRS can be better elucidated by screening target genes and proteins using transcriptome and proteome sequencing, and then performing enrichment and classification according to DEGs and DEPs.

## 1. Introduction

Porcine reproductive and respiratory syndrome virus (PRRSV) is one of the most important causative pathogens of respiratory and reproductive diseases, severely impacting the global swine industry [1]. It is a single-stranded RNA virus. Its viral protein exhibits a high mutation rate, hindering innate immune response [2]. It is listed as a notified disease by the Office International des Epizooties (OIE) and as a type of disease in China [3]. The NADC30-like PRRSV strain is an NA-PRRSV (North American-like) that was isolated in the US in 2008 [4]. The NADC30-like stain has spread throughout China. Many researchers have found that NADC30-like PRRSV led to the re-emergence of new variants with different virulence levels [5,6,7]. It has not only caused the outbreaks of clinical PRRS, but also participated in the recombination and generation of new strains of PRRSV, including recombination with HP-PRRSV (JXA1), classical PRRSV (CH-1a, VR2332), and attenuated vaccines (JXA1-P80, TJM-F92-like) [8,9,10,11].

Additionally, no effective vaccine has been invented for prevention and treatment in recent years. Vaccines sold in China mainly include the inactivated vaccine (CH-1a), classical attenuated vaccines (CH-1R, R98, VR2332), and HP-PRRSV vaccines (JXA1-R, HuN4-F112, TJM-F92, GDr180) [12,13,14,15,16]. Some large-scale pig farms in China are facilitating disease control instead of high-volume vaccination [17,18]. In clinical therapy, most virulent strains are resistant to antiviral drugs. Novel antiviral drugs or compounds are urgently needed to address this problem.

*Isatis* root is a kind of herbaceous plant widely cultivated and distributed in China. *Isatis* root is a joint therapeutic agent of influenza, fever, detoxification, and detumescence in Shennong herbal scripture [19,20]. In recent years, plant polysaccharides have been found to be energy resources that promote animal growth, and participate in animal immune regulation and have antitumor, antibacterial, antiviral, anti-inflammatory effects. *IRPS* has attracted much attention in the field of immunology and virology. Polysaccharides are one of the main components of *Isatis* root. As a macromolecule substance, it exhibits anti-oxidant, anti-inflammatory, and antiviral properties [21,22]. However, the antiviral effect of *IRPS* against PRRSV remains unclear. Previous studies have demonstrated that *IRPS* has substantial anti-oxidant and immunomodulatory activities in vitro, but few studies have been conducted in vivo [23,24]. Furthermore, gut microbes usually degrade polysaccharides through fermentation, producing metabolites including short-chain fatty acids, which participate in the metabolism and affect host physiological status [25,26,27]. Therefore, exploring the effect of polysaccharides on an organism is inseparable from the study of gut microbiota.

Recently, the development and application of “omics” technologies such as genomics, transcriptomics, proteomics, and microbiomics have provided new widely used experimental methods for understanding the molecular mechanisms driving biological processes, cell components, and molecular functions in cells and tissues [28,29,30,31,32,33]. In our study, we explored the antiviral effect of *IRPS* and investigated the expression changes of genes and proteins in three groups of piglets (mock group (M), prevention group (P) and infection group (I)) using transcriptome and proteome sequencing. Some key differentially expressed genes (DEGs), such as immune pathways, chemokine signaling pathways, and metabolic pathways, have highlighted the antiviral effects of *IRPS*. Furthermore, we evaluated the microbiomics in the above three groups of piglets. Our comparative analysis will help the better understanding of the antiviral effect of *IRPS* and develop novel antiviral therapies to provide a scientific reference for the prevention and treatment of PRRSV in the swine industry.

## 2. Results

### 2.1. Toxicity of IRPS on the Viability of Marc-145 Cells

The MTT assay confirmed that the concentration of *IRPS* used in subsequent experiments did not affect cell viability. The tested concentrations of *IRPS* were 0.447, 0.224, 0.112, 0.056, and 0.028 mg/mL. The results showed that *IRPS* had a cytotoxic effect when its concentration was higher than 0.224 mg/mL. Thus, *IRPS* concentrations of 0.027–0.112 mg/mL were selected for the next experiment.

### 2.2. Antiviral Effect of IRPS on NADC30-like PRRSV

The preventive and inhibitory effects of IRPS on PRRSV were investigated by MTT assays. When IRPS concentration increased from 3.5 to 112 μg/mL, it could prevent PRRSV infection.

To evaluate the effect of IRPS on PRRSV replication, Marc-145 cells were infected with 100 TCID50 of PRRSV and were supplied with various concentrations of IRPS. The results showed that the inhibitory effect of IRPS on PRRSV replication was inconspicuous.

There was significant difference in OD570 between the 112 μg/mL of IRPS and 0 μg/mL of IRPS (*p* < 0.05) in inhibitory (Figure 1A) and direct groups (*p* < 0.01) (Figure 1C), respectively. When IRPS concentration increased from 56 to 112 μg/mL, the inhibition percentage increased in the inhibitory and direct groups (Figure 1D). In addition, we found that the IRPS prevented the PRRSV infection in a dose-dependent manner. We found significant differences in OD570 absorbance between 112 μg/mL and 0 μg/mL IRPS (*p* < 0.0001) (Figure 1B). The results showed that IRPS has clinical research potential to prevent PRRSV infection.

### 2.3. Clinical Signs and Histopathology

At 3 dpi, piglets in the infection group exhibited clinical signs of feed intake decreasing from 0.75 kg to 0.52 kg (per piglet). At 4 dpi, clinical signs improved in these piglets, such as messy fur, hypokinesia, anorexia, depression, and fever, and rectal temperatures increased by 0.2 °C to 0.5 °C. During the trial, group P and group M showed no clinical signs. The lungs were separated from the thoracic cavity and pathologically assessed by gross examination, H&E staining, and immunohistochemistry. The lungs exhibited serious hemorrhagic spots, congestion, and emphysema in group I. However, the typical lesions were slightly hemorrhagic spots on the surface of the lungs from group P piglets after being challenged with NADC30-like PRRSV strain, indicating the protective effect of *IRPS* on target organs.

To observe the histopathological changes in piglets, lung samples were fixed with formalin, embedded in paraffin, sectioned, and stained for further analysis. Analysis showed that the alveolar wall was seriously thickened in group I, and there was a small number of inflammatory cells in the pulmonary interstitium (Figure 1E). Some areas exhibited vascular smooth muscle cell necrosis, nucleolysis disappeared, and there was eosinophilic and homogenous in group P (Figure 1E). There was no significant pathological phenotype in the lungs of piglets in group M (Figure 1E). The results revealed that *IRPS* could significantly prevent and efficiently inhibit virus infection. Moreover, it exhibited a significant potential to improve the pathological injury of pig lung tissue.

### 2.4. Overview of Transcriptomic Analyses

In the transcriptomic analysis, 371 million clean reads and 95.93% clean reads were located on the Sus scrofa genome sequence. The Q30 base rate ranged from 92.84 to 93.45%, and 38.8–44.3 million clean reads were obtained per sample (Appendix A). A total of 114,753 genes were detected in three groups by fragments per kilobase million (FPKM) mapped fragments. FPKM > 1 was used as the expression genes. A total of 1669 DEGs were identified in three groups using a cutoff of 2-fold change, FPKM > 1, and *p*-value < 0.05 [34]. There was a total of 1669 DEGs; 917 DEGs had higher expression levels and 752 DEGs displayed opposing tendencies. Among them, there were 870 (482 up-regulated and 388 down-regulated) in M vs. I (Figure 2A), 749 (417 up-regulated and 332 down-regulated) in P vs. I (Figure 2B), 50 (18 up-regulated and 32 down-regulated) in P vs. M (Figure 2C). More in group P and group M than group I, C-X-C motif chemokine ligand 10 (CXCL10), membrane-spanning 4-domains A7 (MS4A7), and interferon regulatory factor 7 (IRF7) had large values with |log2 fold changes| > 2. Meanwhile, group P and M were clustered together first and significantly separated from the group I (Figure 2D). The group M, group P, and group I samples shared 7 DEGs (Figure 2E).

As shown in Figure 2F, the expression trends for 20 genes in the three groups were consistent with the transcriptome analysis (Appendix A). Meanwhile, qRT-PCR results showed that the transcriptome sequencing was reliable.

### 2.5. Gene Ontology Analysis and KEGG Pathway of DEGs

To further use DEG information, GO and KEGG analyses were used to identify relevant biological functions and pathways. As shown in Appendix A, the mitotic cell cycle was the first place in M vs. I, defense response was the first place in P vs. I, and positive regulation of long-term synaptic potentiation was the first place in P vs. M in the biological process (BP). In molecular function (MF), carbohydrate derivative binding accounted for the largest proportion in M vs. I, chemokine activity occupied the largest proportion in P vs. I, and active transmembrane transporter activity occupied the largest proportion in P vs. M. In cellular components (CC), the chromosome and centromeric region were the largest category in M vs. I, the extracellular region was the largest category in P vs. I, and basement membrane was the largest category in P vs. M.

We compared these DEGs to investigate the known immune and signaling pathways to identify the related biological functions and pathways. There were 101 KEGG pathways differentially expressed in the P vs. M samples, 261 KEGG in the P vs. I, and 261 KEGG in the M vs. I, respectively. As shown in Appendix A, the top 20 KEGG pathways were screened based on the number of DEGs. Three pairwise comparisons (M vs. I, P vs. I, and P vs. M) involved cell growth and death, the immune system, signaling molecules, and interaction pathways. The specific metabolic pathways are as follows: cytokine–cytokine receptor interaction, cell cycle, DNA replication, and chemokine signaling pathway, toll-like/NOD-like receptor-signaling pathway, cell adhesion molecules (CAMs), and long-term potentiation.

### 2.6. Proteomic Analysis and Identification of Differentially Expressed Proteins

There were 39,045 identified peptide-spectrum matches (PSM) that matched 18,343 peptides and 4398 protein groups. The EDPs were screened between P and I, between M and I, and between P and M was perfomed by proteomics. Using a threshold of 1.5-fold change and *p* < 0.05 to define DEPs, a total 370 DEPs (195 up-regulated and 175 down-regulated) were detected in the three group samples. Among these DEPs, there were 89 DEPs (38 up-regulated and 51 down-regulated) in the P vs. I, 135 DEPs (64 up-regulated and 71 down-regulated) in the M vs. I, and 146 DEPs (93 up-regulated and 53 down-regulated) in the P vs. M (Appendix A).

As shown in Figure 3A, in BP, the immune-system process was first place in M vs. I, the multi-organism process was first place in P vs. I, and cellular component organization or biogenesis was first place in P vs. M. In MF, chemorepellent activity accounted for the largest proportion in M vs. I, binding occupied the largest proportion in P vs. I, and molecular function regulator occupied the largest proportion in P vs. M. In CC, the chromosome and centromeric region was the foremost category in M vs. I, binding was the foremost category in P vs. I, and supramolecular fiber was the foremost category in P vs. M.

KEGG functional enrichment analysis of the DEPs revealed that primary immune-related pathways, such as herpes simplex infection and measles, were enriched in M vs. I and P vs. I, antigen processing and presentation was enriched in M vs. I and P vs. M, and influenza A was enriched in M vs. I and P vs. M. It is noteworthy that the PPAR signaling pathway and rheumatoid arthritis were enriched in M vs. I, tight junction and mineral absorption were enriched in P vs. I, and oxidative phosphorylation, fatty acid degradation, and butanoate metabolism were enriched in P vs. M (Figure 3B).

### 2.7. Correlation Analysis of DEGs and DEPs

In M vs. I, P vs. I, and P vs. M, the R values of the correlation coefficient were 0.387, 0.412, and 0.055 between proteome and transcriptome (Figure 4A–C). To correlate transcript and protein expression profiles, accession numbers were extracted from the proteome and compared with annotated RNA-Seq libraries, which found 33 proteins corresponding to DEPs and DEGs (Appendix A). A total of 30 of the 174 proteins had the same expression levels as mRNA levels, including 19 proteins from M vs. I, and 11 proteins from P vs. I. The 33 DEGs and DEPs were conducted by cluster analyses (Figure 4D). Then, 30 genes showed similar regulation, and 3 genes showed opposite regulation, after comparison at transcript and protein levels.

### 2.8. Effects of IRPS on Gut Microbiota Diversity

#### 2.8.1. The Microbial Community Diversity of Piglets

In this study, the number of effective sequences in each sample exceeded 75,000. The length distribution of piglet samples was about 400–450 bp. Figure 4 shows the ASVs of each piglet at Phylum, Class, Order, Family, Genus, and Species levels. Over 220 ASVs have been identified per sample at the Species level (Figure 5A). The Venn diagram reveals 710 bacterial species in all three groups. Interestingly, there were 282 bacterial species in group I and group P, but not found in group M (Figure 5B).

#### 2.8.2. Alpha Diversity and Beta Diversity

In this study, seven indices were used to represent community diversity. There were no significant changes among groups M, P, and I (*p* > 0.05) (Figure 5C). Principal coordinate analysis (PCoA) can reflect the similarity between samples according to the projected distance of samples on the coordinate axis [35]. Using weighted UniFrac distance-based PCoA, Figure 5D showed that the projection distance of group I was the farthest from group M and P, while group P was the closest to group M, indicating that *IRPS* had an improved effect on the microbiota composition of infected piglets.

#### 2.8.3. Gut Microbiota Community Structure Analysis

At the Phylum level, firmicutes (74.27% in group P, 51.18% in group I, 68.72% in group M), bacteroidetes (17.75% in group P, 41.46% in group I, 17.64% in group M), and spirochaetes (5.19% in group P, 4.14% in group I, 10.93% in group M) were found to be the major microbial community of all piglets (Appendix A). At the Genus level, streptococcus was found to be much higher in group M (35.1%) and group P (29.95%) than group I (0.27%), while prevotella was found to be much higher in group I (27.63%) than group P (6.33%) and group M (7.15%). Treponema was found to be much higher in group M (10.52%) than group P (4.96%) and group I (3.86%). There was no visible difference among different groups of oscillopsia. Roseburia was found to be much higher in group I (8.9%) than group P (0.09%) and group M (0.1%) (Appendix A).

#### 2.8.4. Comparison of the Microbial Community Structure among Different Piglet Groups

At the Phylum level, the dominant bacteria were composed of firmicutes, bacteroidetes, and spirochaetes in each group. Among them, firmicutes abundance was highest in the three groups, and *IRPS* had a significant effect on it (*p* < 0.05). Compared to group I, group P was characterized by higher firmicutes levels. The relative abundance of bacteroidetes in groups P and M was significantly lower than group I (*p* < 0.05) (Figure 6A). At the Genus level, compared to the group I, streptococcus, roseburia, and phascolarctobacterium were significantly different, with *p* < 0.05 and prevotella (*p* < 0.01) in group P. However, compared to group I, streptococcus and roseburia were significantly different at *p* < 0.05, while prevotella, SMB53, and phascolarctobacterium were extremely significant different at *p* < 0.01 in group M (Figure 6C). Meanwhile, we found a difference in gemmiger and anaerovibrio between group I and other groups using heat map analysis (Figure 6B).

## 3. Discussion

In this study, *IRPS* was proved to prevent and efficiently inhibit virus infection in vitro and in vivo. In addition, the objective was to explore the potential immune mechanism of *IRPS* involvement in the antiviral response of piglets. There were analyses that identified many DEGs and DEPs using transcriptome and proteome sequencing. In addition, the changes in intestinal flora were observed by 16S sequencing.

*Isatis* root, a traditional Chinese medicine (TCM) known for its widely antiviral effects, has a medical history of thousands of years in China and other Asian countries [21]. It has been reported that *IRPS* has antiviral effects on swine influenza virus (SIV), herpes simplex virus 1 (HSV-1), pseudorabies virus (PRV), and severe acute respiratory syndrome (SARS). We compared and integrated the antiviral effects of *IRPS* in vitro (preventive effect, inhibitory effect, and direct effect). The results showed that *IRPS* has a preventive effect. Subsequently, the preventive effect of *IRPS* was demonstrated in vivo. Interestingly, it exhibited significant potential to improve the pathological injury of pig lung tissue.

In this investigation, the DEGs and DEPs of SLA-2, CD180, ANXA13, MX2, CD14, LOC110261034, and DHX58 were found in group I and group P (as also observed in the study by Lim [34]). Furthermore, DDX58, DCN, CD163, AK1, PCNA, KIF23, CTSB, OAS2, and GZMA were found in group I, but not in other groups (Appendix A). DDX58, OAS2, and CXCL10 were highly expressed in 3 dpi. OAS2 is one of the 2–5A synthetase families, which promote the degradation of viral RNA and the inhibition of viral replication [36]. CD163 is a receptor required for PRRSV infection and its expression is regulated by a variety of inflammatory mediators. Several studies have shown that CD163 is an essential receptor involved in the uncoating of virions, blocking or down-regulating CD163 expression to restrict PRRSV infection [37,38,39,40]. In our study, there was no significant change in mRNA and protein levels of CD163 between P and I, suggesting that CD163 was more likely to be related to an *IRPS* antiviral effect. Moreover, *IRPS* may reduce PRRSV infection by inhibiting or blocking the expression of CD163.

PRRSV may be similar to other RNA viruses or coronaviruses. Coronavirus is a single-stranded RNA virus. PRRSV produces dsRNA early during the infection cycle due to genome replication and mRNA transcription [41,42]. The DEGs/DEPs could be mapped to the immune pathways of piglets. Among them, the DEGs/DEPs were mapped onto the NOD-like receptor-signaling pathway using the KEGG database. Oligoadenylate synthetase (OAS), NLR Family Pyrin Domain Containing 3 (NLRP3), Caspase 1 (CASP1), and Interleukin 18 (IL-18) were significantly up-regulated in group I vs. group M. Interleukin-1 beta (IL-1β) showed no significant changes. Previous studies have shown that the non-structural protein 11 of PRRSV inhibits NLRP3 inflammasome-mediated induction of IL-1β [43]. On the other hand, PRRSV could induce early IL-1β expression and secretion in porcine alveolar macrophages (PAMs) via the NLRP3 inflammasome pathway. However, the levels of pro-IL-1β mRNA and IL-1β protein decreased to a certain extent after infection [44].

NLRP3 only changed in group I, and no significant change was found in other groups. The *IRPS* was secure for the host and non-activate inflammatory pathways. NLRP3 inflammasome induces maturity and secretion of proinflammatory IL-1β and IL-18 as a multiprotein complex. The maturation of IL-1 and IL-18 leads to signaling cascades that regulate the expression of tumor necrosis factor alpha (TNF-α), interferon gamma (IFN-γ), and other cytokines; these recruit the lymphocytes, linking innate and adaptive immune responses to control the invading pathogens [45,46,47].

In the past decade, numerous studies have attributed to the prevotella species. They have been abundant under inflammatory conditions, such as rheumatoid arthritis (RA), periodontitis, intestinal dysbiosis, and inflammation in HIV patients. The previous report indicated that increased prevotella in human immunodeficiency virus (HIV) is a driver of persistent gut inflammation, leading to mucosal dysfunction and systemic inflammation [48,49,50]. *IRPS* could inhibit prevotella and reduce inflammatory response in our study. Furthermore, the relative abundance of streptococcus also returned to normal levels.

These results suggest that *IRPS* not only inhibits inflammatory gene expression, but also reduces the inflammatory response of hosts. In a previous study, astragalus polysaccharide (APS) was found to alleviate colitis in mice by inhibiting NLRP3 inflammasome [51]. Thus, the inhibitory effect of *IRPS* on PRRSV-induced inflammation may be due to the inhibition of the NLRP3 gene expression, and affect the expression of downstream genes. The antiviral mechanism of *IRPS* against PRRSV is summarized in Figure 7. In summary, we report here provide a molecular rationale to explain the broad antiviral properties of *IRPS*.

## 4. Materials and Methods

### 4.1. Preparation of IRPS

*Isatis* root was purchased from Beijing Shengtaier Biological Technology Co., Ltd. (Beijing, China). The preparation of *IRPS* from *Isatis* root was performed by the phenol-sulfuric acid method. Briefly, *Isatis* root crude extracts were immersed in deionization water for dissolution. Subsequently, 1 mL solution was mixed with 1.0 mL of 6% phenol solution. After being mixed well, 5.0 mL of concentrated sulfuric acid was added and mixed immediately. The solvent was heated at 70 °C for 20 min and stored at 16 °C. The absorbance values of OD570 were measured using an automated plate reader (Bio-Rad, Hercules, CA, USA). All samples were prepared in triplicate. The final concentration of *Isatis* root polysaccharide was 0.447 mg/mL.

### 4.2. Virus and Cells

NADC30-like PRRSV strain used in this study was isolated from a farmed pig in Sichuan province, China, in 2016. In short, the representative clinical symptoms include persistent high fever, anorexia, blue ears, and conjunctivitis. Clinical specimens (lung, liver, lymph) were collected for detection, and the virus was identified as the NADC30-like PRRSV strain by PCR and sequencing techniques. In the previous experiments, 50% tissue culture infection dose (TCID50) of the 10^4.36^·100 μL^−1^ was calculated by the Reed–Muench method.

Monkey kidney (Marc-145) cell line was maintained at the college of veterinary medicine (Sichuan Agricultural University, China). According to the standard culturing procedure, cells were cultured with DMEM containing 10% fetal bovine serum, 100 U/mL penicillin, and 100 U/mL streptomycin at 37 °C with 5% CO_2_.

### 4.3. Animal Protocol and Statement

Fifteen male piglets (three-breed cross; 5–6 weeks old; 10–13 kg) were purchased from Chengdu Wangjiang agriculture and animal husbandry technology Co., Ltd. (Chengdu, China). According to qPCR diagnosis, all piglets have not been vaccinated or infected with common porcine pathogens. All animal experiments were approved by the institutional animal care and use committee of Sichuan Agricultural University of China (Approval number SYXK2019-187). All animal experiments were performed under Laboratory Animals Welfare and Ethics published by the General Administration of Quality Supervision, Inspection and Quarantine of the People’s Republic of China. All animals were individually housed under controlled temperature (26 °C), humidity (60%), and lighting (12 h/day), with free access to water.

### 4.4. Anti-NADC30-like PRRSV Activity In Vitro

#### 4.4.1. Determination of IRPS Cytotoxicity

The methyl thiazolyl tetrazolium (MTT) assay was used to evaluate the cytotoxicity effect of *IRPS* in Marc-145 cells [23,52]. Marc-145 cells were inoculated into 96-well culture plates with 2.4 × 10^4^ cells/well. After 36 h of culture, different concentrations of *IRPS* were added to each well and maintained for 120 h in triplicate. Then, MTT reagent was supplied and incubated at 37 °C with 5% CO_2_ for 4 h followed by adding 100 μL of DMSO (Gibco, Grand Island, NY, USA) in each well. The optical densities (OD) values were measured at the wavelength of 570 nm using an automated plate reader (Bio-Rad, Hercules, CA, USA).

#### 4.4.2. Determination of IRPS Cytotoxicity

The MTT assay was used to evaluate the preventive effect of *IRPS* on PRRSV replication [53]. Marc-145 cells were cultured for 24 h and reached at least 80% confluence in 96-well plates. Then, the culture medium was refreshed in each well with various concentrations of *IRPS*. After incubation for 4 h, *IRPS* was washed with PBS three times for removal, and the cells were infected with PRRSV of 100 TCID50 for 1.5 h. Subsequently, the non-absorbed viral particles were discarded by being washed with PBS three times and supplied with fresh culture medium. The cells were then incubated at 37 °C with 5% CO_2_. The control group cells (non-added *IRPS*) were incubated to reach at least 80% cytotoxicity. The infection percentage of PRRSV was determined by the following formula: virus inhibition rate (%) = (ODt − ODv)/(ODc − ODv) × 100, in which the ODt, ODc, and ODv represent the absorbance of tested substances (cells, virus, *IRPS*), cell control (cells) and virus control (cells, virus) [24].

Marc-145 cells were cultured in 96-well plates, as described. When the cell culture reached at least 80% confluence for 24 h, the medium was removed, and 100 TCID50 of PRRSV was added to each well. After incubation for 1.5 h, the virus was washed with PBS three times to remove the unabsorbed virus. Then, the various concentrations of *IRPS* were added to each well. The MTT assay was used to evaluate the inhibitory effect of *IRPS* as previously mentioned.

Briefly, several different concentrations of *IRPS* were thoroughly mixed with PRRSV of 100 TCID50 and incubated at 37 °C for 1 h. Then, the mixture was added to cells with at least 80% confluence. After incubation for 1.5 h to observe cell status, the MTT assay was used to evaluate the direct effect of *IRPS* on PRRSV by the above method.

Statistical analysis was performed with GraphPad Prism 8.0.2 (GraphPad Software, San Diego, CA, USA) using the *t*-test.

#### 4.4.3. Antiviral Effect of IRPS In Vivo

After 7 d of domestication, 15 piglets were randomly divided into three groups (n = 5). Group 1 was the infection group. Group 2 was the prevention group. Group 3 was the mock group. Group 2 was treated by feeding with doses of *Isatis* root granule (0.1 g/kg/d).

After 15 days of feeding, all piglets were slightly anesthetized and intranasally inoculated with 105.38 TCID50 of NADC30-like PRRSV, except group 3, who were inoculated with culture medium. Then, oral and nasal samples were collected at 5 days post-challenge for the detection of the PRRSV NADV30-like genomic RNA by qPCR. All piglets were observed to record clinical signs, such as mental state, breathing, and eating habits. Any piglet deaths were recorded during the PRRSV challenge, and they were euthanized on day 15. In addition, lung tissue samples were collected and fixed in 4% buffer formaldehyde solution for 1 d. They were embedded in paraffin, sectioned, and stained with eosin. Subsequently, the slides were observed under a microscope.

### 4.5. RNA Extraction and Sequencing

The lung tissue samples were collected from three groups of piglets (group I: infection group; group P: prevention group; group M: mock group.). Total RNA was extracted from lung tissues using the TRIzol reagent according to the manufacturer’s protocol. The concentration and quality of RNA were evaluated by NanoDrop 2000 Spectrophotometer and Agilent 2100 Bioanalyzer. The cDNA libraries were constructed using only high-quality RNA (RNA integrity number, RIN > 7.0). All libraries were sequenced on a HiSeq 4000 with paired end (PE, 350 bp). By removing reads containing adapters, low-quality reads, and reads containing more than 5% N (default parameters), the obtained raw data were filtered into clean data with FASTP, and then mapped to the pig reference genome of Sus scrofa 11.1 using HISAT2 [54,55]. The gene expression levels of fragments per kilobase million (FPKM) were analyzed. Absolute fold change (|log2 fold change|) > 1 and *p*-value < 0.05 were used as the thresholds for screening DEGs [56]. We used R-package ClusterProfiler for Gene Ontology (GO) enrichment to analysis DEGs. The statistical enrichment of DEGs analysis was conducted using R software according to the Kyoto Encyclopedia of Genes and Genomes (KEGG) pathways [32,57,58].

### 4.6. Quantitative Real-Time PCR Validation

Total RNA was extracted from the lung tissue of nine piglets (group I, group P, group M) for qRT-PCR analysis. The expressions of 20 swine-related genes (12 up-regulated and 8 down-regulated) were verified by quantitative real-time PCR (qRT-PCR). All experiments were carried out 3 times. The expression levels of internal reference gene were normalized to β-Actin levels. The qRT-PCR reaction was conducted as follows: 95 °C for 3 min, a total of 40 cycles of 95 °C 10 s, 59 °C 10 s, and 72 °C 30 s. The relative gene expression levels were evaluated by 2^−ΔΔCt^ method [59].

### 4.7. Tandem Mass Tag-Based on Proteomics

Differentially expressed proteins (DEPs) were detected in the piglets with divergent lung traits by isobaric tandem mass tags (TMTs). Each peptide was about 80 μg. The total protein (100 μg) was removed from each sample solution and digested with trypsin (Thermo Fisher Scientific, Waltham, CA, USA) at 37 °C for 16 h at a protein:trypsin ratio of 30:1. After digestion, the peptides were dried by vacuum centrifugation. TMT labeling was performed on ten-plex kit (Thermo Fisher Scientific, San Jose, CA, USA) according to the manufacturer’s instructions. Nine samples (3 biological replicates from the Mock group, 3 biological replicates from the infection group, 3 biological replicates from the prevention group) were TMT labeled: 126-, 127C-, and 128N-TMT tags for the control group; 128C-, 129N-, and 129C-TMT tags for infection group; 130N-, 130C-, and 131- for the prevention group. Then, the labeled samples were analyzed using liquid chromatography-tandem mass spectrometry (LC-MS/MS) [60]. Based on a previous study, the Easy-nLCTM 1200 ultra-high-performance liquid chromatography (UHPLC) system (Thermo Fisher Scientific, Waltham, MA, USA) was used for strong cation exchange (SCX) chromatography [61]. The peptide was analyzed by Q-Exactive mass spectrometer, positive ion mode, and the selected mass range was 300–1800 mass/charge (m/z).

The raw files were first merged and converted into an MGF file by Proteome Discoverer 1.4 (Thermo Fisher Scientific, San Jose, CA, USA). Then, a search was performed on the Uniprot database of swine (Sus scrofa) Mascot 2.2 search engine (Matrix Science, London, UK) with 180,744 entries. The relevant parameters are as follows (Table 1). Protein expression levels with a cutoff of 1.5-fold change and *p*-value <  0.05 were determined as significant DEPs. For bioinformatics analysis, the parameter sets were the same as those in the study of Wang et al. [31].

Using QuickGO (http://www.ebi.ac.uk/QuickGO/) (accessed on 21 June 2021) Fisher’s exact test evaluated the significance level of the GO term after DEP enrichment, and considered *p*-value < 0.05 as significant. According to the Kyoto Encyclopedia of Genes and Genomes (KEGG) pathway database (https://www.kegg.jp/kegg) (accessed on 27 June 2021), metabolic pathway analysis was conducted using R software, version 3.5.1(used on 2 August 2021).

### 4.8. DNA Extraction of Intestinal Contents and Sequencing

Fresh feces samples were collected from the infection group (n = 3; marked as I1, I2, I3), prevention group (n = 3; marked as P1, P2, P3), mock group (n = 3; marked as M1, M2, M3). According to the manufacturer’s directions, total microbial genomic DNA was extracted from three groups of samples using the QIAam DNA Mini Kit (Qiagen, Hilden, Germany). The PCR was amplified the V3–V4 region of 16S rRNA genes and sequenced on the Illumina MiSeq/NovaSeq platform.

The incorrect or error sequences were removed by Quantitative Insights into Microbial Ecology, v.2019 (QIIME2) [62]. QIIME2 employs the Divisive Amplicon Denoising Algorithm (DADA2) to identify amplicon sequence variants (ASVs). DADA2 is an open-source software package for modeling and correcting amplicon errors in Illumina sequencing. DADA2 accurately infers sample sequences without coarse graining into OTUs, and resolves only one nucleotide [63,64]. Then, ASVs were taxonomically grouped and classified by plotting classify-sklearn (https://github.com/QIIME2/q2-feature-classifier) (accessed on 14 July 2021) against a curated Greengenes database (Release 13.8, http://greengenes.secondgenome.com/) (accessed on 20 July 2021) [65,66]. Seven indices of alpha diversity were computed (Chao1and Observed species, Faith’s PD, Simpson and Shannon, Pielou’s evenness, Good’s coverage) [67,68,69,70,71,72,73]. Principal component analysis was used to analyze the beta diversity of microbial community structure similarity [74]. QIIME2 was used for statistics of microbial community structure in Phylum, Class, Order, Family, Genus, and Species of each group.

For statistical analysis, the parameters were set the same as in the study by Chen et al. [27]. All data were analyzed by one-way ANOVA, followed by multiple comparisons using the Tukey test.

## 5. Conclusions

In conclusion, this study was the first attempt to investigate the complexity of *IRPS* antiviral effects against PRRSV at the transcriptome and proteome levels of piglets. The interpretation of two omics data revealed many candidate genes and proteins that may be involved in the antiviral effect of *IRPS* against PRRSV mechanisms of the piglet. These DEGs or DEPs were involved in the immune-system and infectious-disease (viral) pathways, such as toll-like and Nod-like receptor-signaling pathway, Influenza A, herpes simplex infection, and chemokine signaling pathway. Further studies demonstrated that the antiviral effect of *IRPS* might be related to the Nod-like signaling pathway, which mainly reduces the inflammatory response by inhibiting the NLRP3 caspase-1 gene expression. Our findings will facilitate further research aimed at identifying essential genes and proteins active during the antiviral activity of *IRPS* against PRRSV. However, more work is needed to clarify the specific functions of these identified genes and proteins. Furthermore, this study demonstrated that *IRPS* might impact PRRSV infection in piglets by intervention or modulation of gut microbiota. Moreover, *IRPS* could restore the microbial community richness of infected piglets and has a potentially beneficial effect on the host by mediating the structure of gut microbiota.

## Figures and Tables

**Figure 1 ijms-23-03688-f001:**
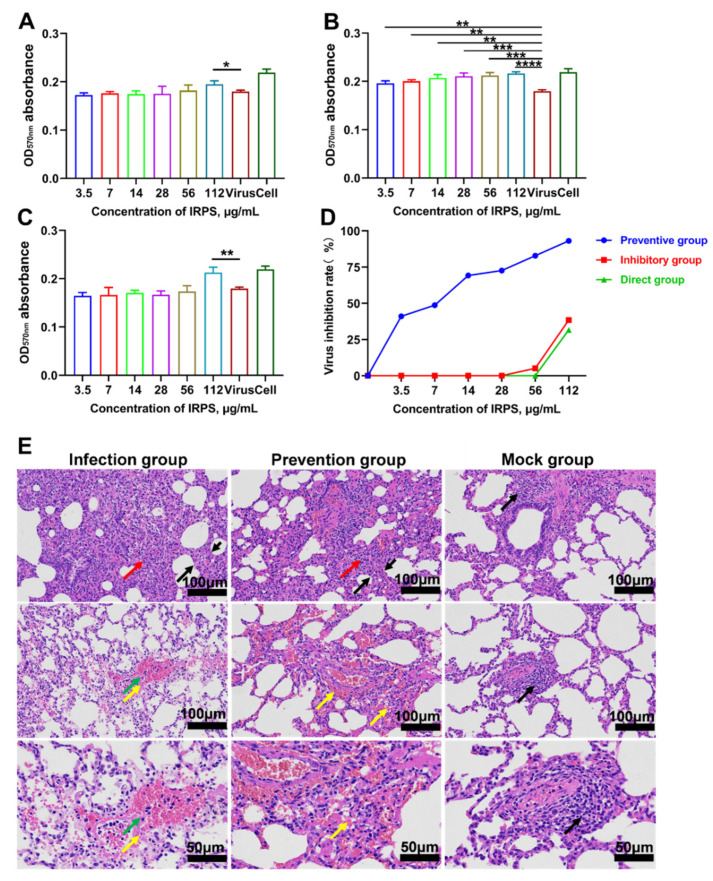
Antiviral study of *IRPS* in vivo and in vitro. (**A**) Inhibitory effect of *IRPS* on PRRSV; in the inhibitory assay, at 1.5 h PRRSV infection, cells were treated with different concentrations of *IRPS* and cell status was observed; (**B**) Preventive effect of *IRPS* on PRRSV; in the preventive assay, cells were treated with different concentrations of *IRPS* for 4 h, then 1.5 h PRRSV infection to observe cell status; (**C**) The direct effect of *IRPS* on PRRSV; in the direct assay, PRRSV was mixed with various concentrations of IRPS and incubated for 1.5 h to observe cell status; (**D**) Percentage inhibition of *IRPS* on PRRSV; Virus: the concentration of *IRPS* was 0 μg/mL. Cell: the normal growing cell. * (*p* < 0.05), ** (*p* < 0.01), *** (*p* < 0.001), **** (*p* < 0.0001). (**E**) Effect of *IRPS* on lung histopathology of PRRSV-infected piglets. Histological observation of representative lungs from three groups (H&E stain, 100× and 50× magnification). Infection group: severe thickening of alveolar walls in local tissue(black arrow); alveolar cells can be observed in a small inflammatory cell infiltration (red arrow). local vascular smooth muscle cell necrosis, nuclear dissolution disappears, and eosinophilic homogeneous (yellow arrow); surrounding bleeding (green arrow). Prevention group: local alveolar wall thickening of tissue (black arrow); alveolar wall can be seen in a small inflammatory cell infiltration (red arrow); local hemorrhage, alveolar cavity, and blood vessels can be seen with more red blood cells (yellow arrow). Normal control group: lung tissue of blank group. Tissue alveolar; the bronchial structure is clear; thin alveolar wall; and small inflammatory cell infiltration (black arrow) around bronchial and blood vessels (black arrow); the branch structure is complete, and the epithelial cell morphology is normal.

**Figure 2 ijms-23-03688-f002:**
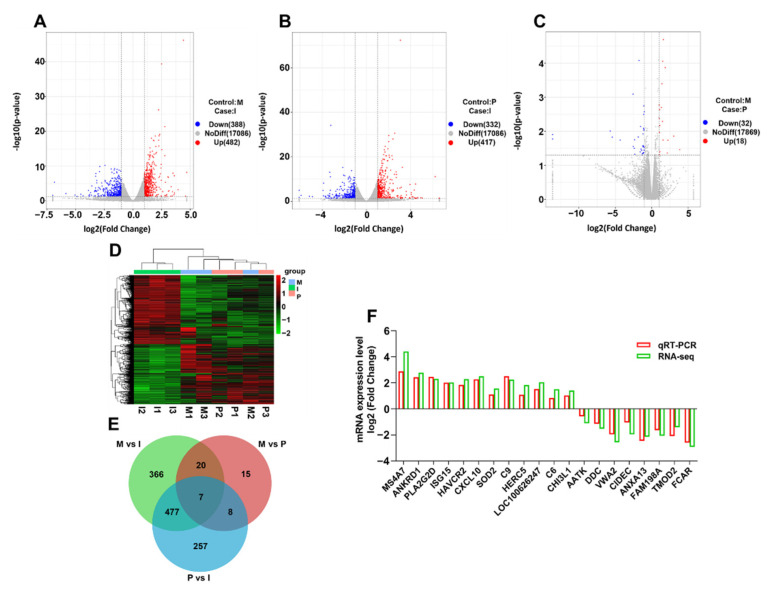
Transcriptome differences between the piglet lung tissue samples. Plot showing the log2 fold change and the −log10(*p*-value). The red points represented the up-regulated DEGs, and the green points represented the down-regulated DEGs (|log2 fold change| > 1), *p*-value < 0.05). (**A**) group P vs. group I; (**B**) group M vs. group I; (**C**): group M vs. group P; (**D**) Heat map of the DEGs in different groups; (**E**) using the Venn diagram for standard or unique DEGs; (**F**) qRT-PCR analysis of 20 DEG expression changes. The levels of β-actin normalized relative mRNA expression levels.

**Figure 3 ijms-23-03688-f003:**
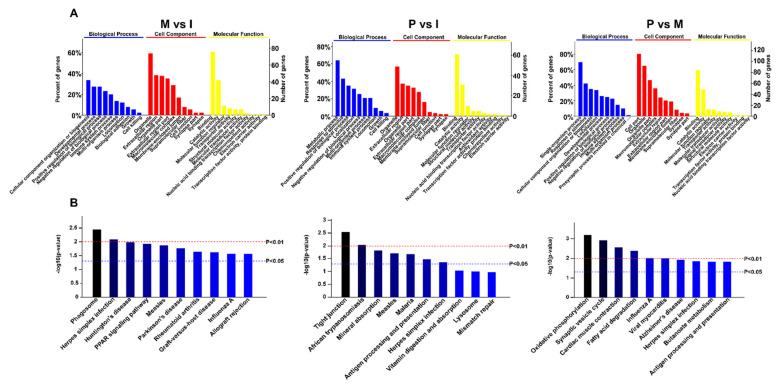
Identification and function analysis of DEPs. (**A**) GO analysis of the DEPs; (**B**) KEGG analysis of the DEPs.

**Figure 4 ijms-23-03688-f004:**
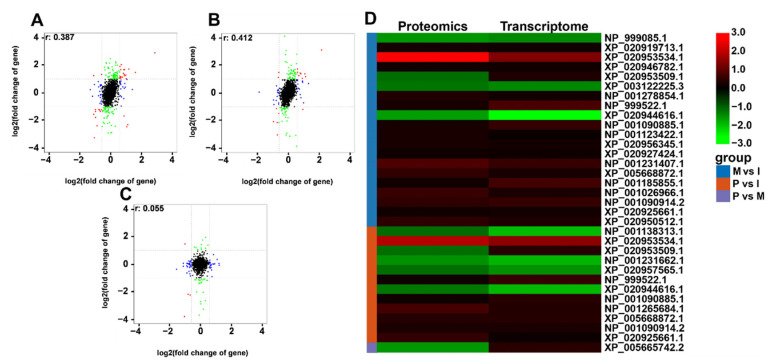
Correlation analysis of proteomics and transcriptome. (**A**–**C**) The correlation analysis of proteomics and transcriptome in M vs. I, P vs. I, P vs. M. In related analysis diagram, DEGs level for absolute fold change (|log2 fold change|) = 1, and the DEPS was absolute fold change (|log2 fold change|) = 0.585 as thresholds for Pearson correlation analysis. Red indicates that the transcript and protein levels are significantly differentially expressed. Blue indicates that the protein levels are significantly differentially expressed. Green indicates that the transcript levels are significantly differentially expressed. Black indicates that the transcript and protein levels are not significantly differentially expressed; (**D**) The expression pattern clustering analysis for 33 DEGs and DEPs with correlations. Red indicates up-regulated, and green indicates down-regulated.

**Figure 5 ijms-23-03688-f005:**
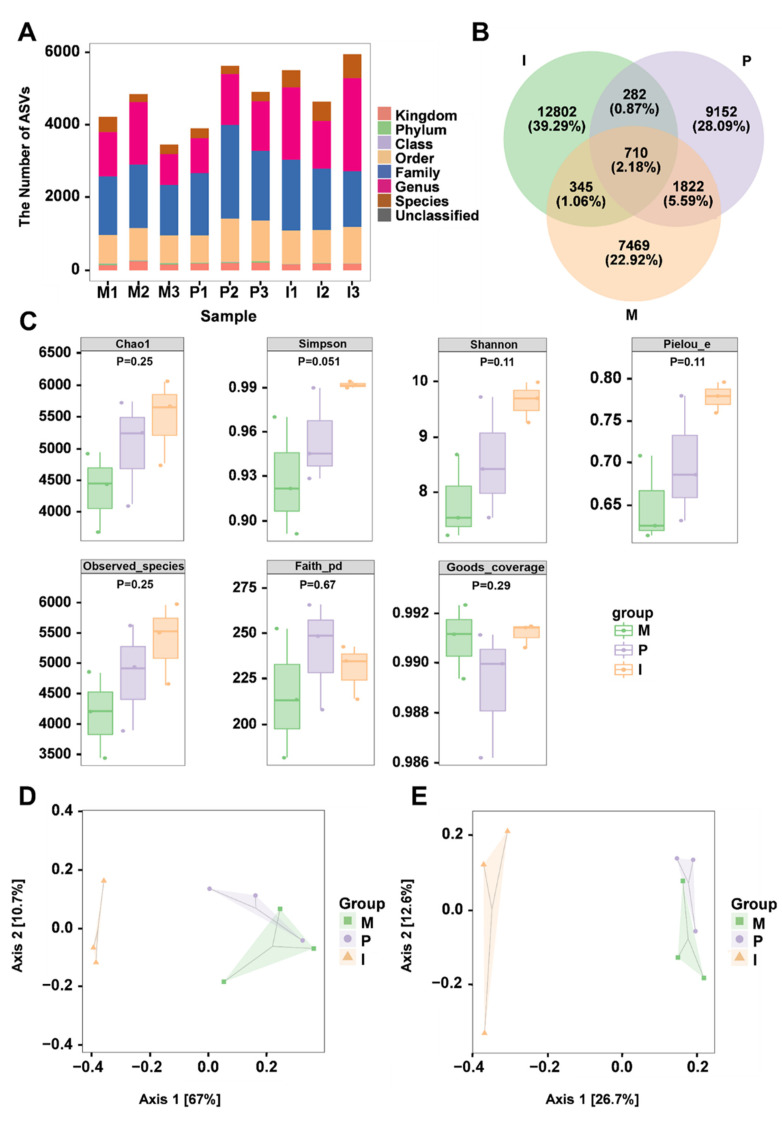
*IRPS* modulated the composition of gut microbiota. (**A**) The quantity of ASVs in different piglet samples; (**B**) Venn diagram of comparison of ASV distribution in different groups; (**C**) Comparison of the microflora microbial diversity index (Chao1, Simpson, Shannon, Pielou’s evenness, Observed species, Faith’s PD) between piglet groups; (**D**) weighted UniFrac distance-based PCoA; (**E**) unweighted UniFrac distance-based PCoA. Axis 1 (X) and 2 (Y) represent the main axes of objects changing in two-dimensional space. Each point in the graph represents a sample, and the dots of different colors indicate different groups of samples. The closer the projection distance of the two points on the coordinate axis, the more similar the community composition of the two samples in the corresponding dimensions.

**Figure 6 ijms-23-03688-f006:**
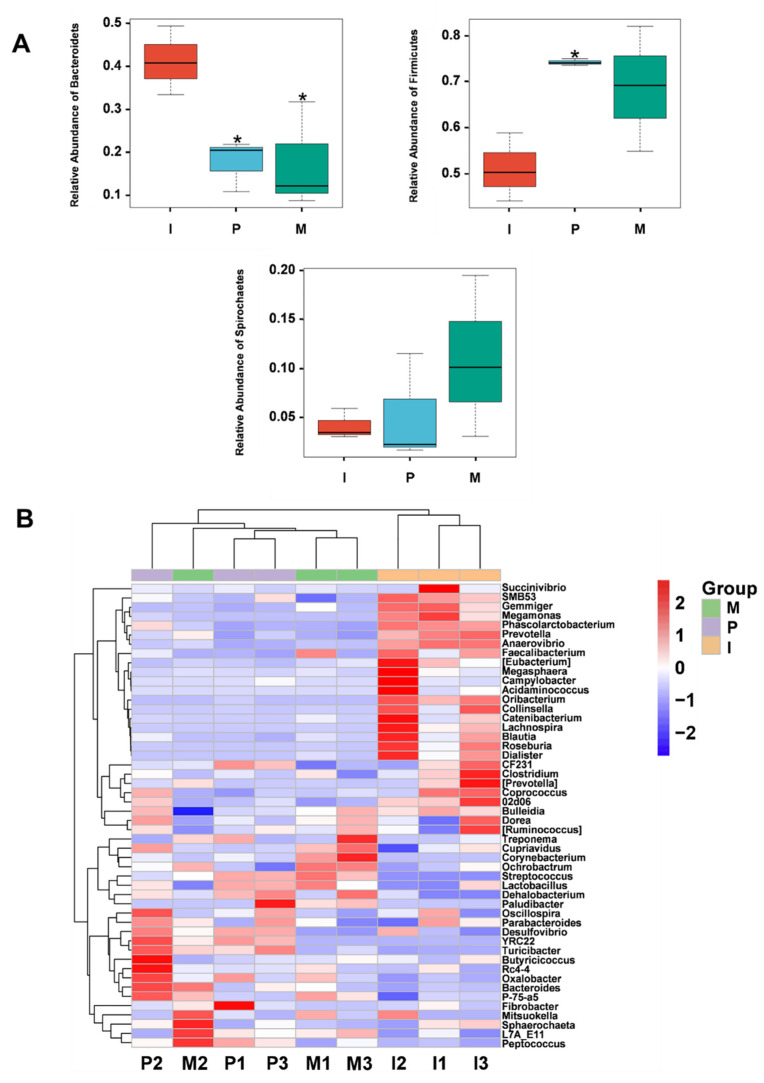
Comparative analysis of gut microbial community structure in different piglet groups. (**A**) Comparison box chart of microbial community structure at the Phylum level; red represents group I, blue represents group P, green represents group M; (**B**) Heat map of the 50 most abundant genera of each group; the red block indicates that the Genus was more abundant in that sample than in other samples, and the blue block indicates that the Genus was less abundant in that sample than in other samples; (**C**) Comparison box chart of microbial community structure at the Genus level. Different asterisks indicate significant differences * (*p* < 0.05), ** (*p* < 0.01).

**Figure 7 ijms-23-03688-f007:**
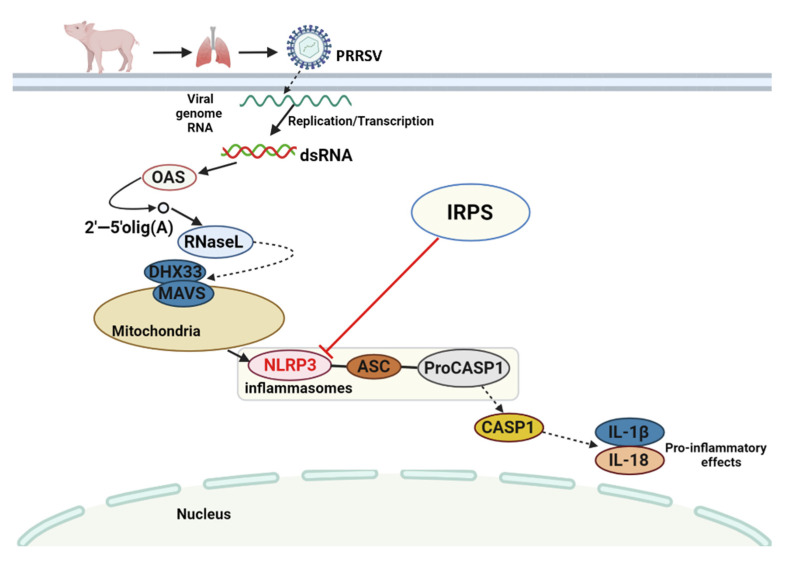
Schematic presentation of the *IRPS* anti-PRRSV molecular mechanism. The graphic was created with BioRender.com. PRRSV becomes double-stranded RNA (dsRNA) by replication and transcription, and acts directly on OAS gene. OAS gene could promote RNaseL expression. RNaseL acts indirectly on DHX33 and MAVS, which in turn activate inflammasomes (NLRP3, ASC, ProCASP1). Then, the inflammasomes act indirectly on CASP1, and finally activates IL-1 and IL-18, thus triggering inflammation in the body. 
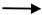
 activation; 
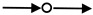
 expression; 
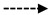
 indirect effect; 
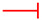
 inhibitory effect.

**Table 1 ijms-23-03688-t001:** Protein samples retrieved; qualitative and quantitative parameters.

Item	Value
Type of search	MS/MS Ion search
Enzyme	Trypsin
Mass Values	Monoisotopic
Max Missed Cleavages	2
Fixed modifications	Cabamidomethyl (C), TMT 10 plex (N-term), TMT 10 plex (K)
Variable modifications	Oxidation (M), TMT 10 plex (Y)
Peptide Mass Tolerance	±20 ppm
Fragment Mass Tolerance	0.1 Da
Protein Mass	Unrestricted
Database	uniprot_Sus_scrofa_180744_20200117.fasta
FDR	≤0.01
Protein Quantification	Use only unique peptides
Experimental Bias	Normalize on protein median

## Data Availability

The data presented in this study are available on request from the corresponding author.

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
