# Peer review of "The Antiviral Effect of Isatis Root Polysaccharide against NADC30-like PRRSV by Transcriptome and Proteome Analysis"

_ijms, 2022, doi:10.3390/ijms23073688_

Round 1

Reviewer 1 Report

This study aimed to investigate the antiviral effects of IRPS against PRRSV and changes in gene expression in proteins in piglets by transcriptome and proteome sequencing. Porcine reproductive and respiratory syndrome virus (PRRSV) is one of the main pathogens that affect the swine industry. Currently, there are no effective vaccines and in clinical therapy, most strains are resistant to the antivirals available on the market. The adoption of herbal medicines extracted from plants has been widely investigated in the pharmaceutical industry, due to their therapeutic properties for the treatment of various diseases. This highlights the importance of the study.

Overall, the study is relevant and original. The introduction is well written and addresses the problem of the study. The material and methods are written clearly and in detail. The results and discussion are appropriate, as well as the conclusion. Of this, considering the quality of the manuscript, I recommend its publication in the International Journal of Molecular Sciences.

Author Response

We would like to thank reviewer for the comments and kindly help. 

Reviewer 2 Report

Jiang et. al., studied antiviral effect of Isatis root by transcriptome and proteome analysis. They identified several genes and pathways corresponding to IRPS antiviral effect. Their study indicated IRPS might act via nod-like signaling pathway. However, there is scope for lot more work in order to identify specific genes and targets of IRPS. The conclusion of study at this point is very broad. The following changes are suggested prior to publication.

Minor English language issues:

Abstract - lines 28-36 needs to be revised e.g. 'IRPS could inhibitor of' 

line 46 - change 'my country to 'China'
line 56 - vaccination 'in' china. Provide references on current vaccines
Lines 308-309 'were found to repetition, study of Lim also observed it'

Other issues:
Line 79 - expand DEG
Figs. 1A-1D need to be more descriptive like Fig. 1E.
Fig. 6 - need more description
ne 328 - provide reference
Figure 7 - needs explanation in text
Lines 351-358 - better fits in previous paragraph before summary.

Author Response

Response to Reviewer Comments

Dear Reviewer,

Thank you for your decision letter concerning our manuscript (ID: ijms-1635391) entitled “ The antiviral effect of Isatis root polysaccharide against NADC30-like PRRSV by transcriptome and proteome analysis ”, and your time regarding for our revision. I also appreciate all the critical comments from you. We have carefully considered the comments and revised the manuscript accordingly. With these improvements, we hope that the current version can meet the Journal’s standards for publication. The following is a point-by-point response to all those comments and a list of changes we have made to the manuscript.

Minor English language issues:

Point 1: Abstract - lines 28-36 needs to be revised e.g. 'IRPS could inhibitor of' 

Response 1: We are very grateful for the comment and suggestion from reviewer #2. We have revised the English language questions. “IRPS could inhibit exert beneficial effects on the host by regulating the structure of intestinal flora. (line 30-36)”.

Point 2: line 46 - change 'my country to 'China'

Response 2: We revised the “my country”to“China”(line 46, 58). “….as a type of disease in China”; ”….volumes of vaccination in China”.

Point 3: line 56 - vaccination 'in' china. Provide references on current vaccines

Response 3: We summarized currently vaccines different vaccines and provided relevant articles (line 55-57). “Vaccines sold in China mainly include inactivated vaccine (CH-1a), classical attenuated vaccines (CH-1R, R98, VR2332), and HP-PRRSV vaccines (JXA1-R, HuN4-F112, TJM-F92, GDr180)”.

Point 4: Lines 308-309 'were found to repetition, study of Lim also observed it'

Response 4: We modified and refined the sentence, “found in group I and group P, study of Lim also observed it” (line 317-318).

Other issues:

Point 1: Line 79 - expand DEG

Response 1:
We have improved the DEG to differentially expressed genes (DEGs). (line 81-82).

Point 2: Figs. 1A-1D need to be more descriptive like Fig. 1E.

Response 2: We explained and supplemented the relevant contents of Figure 1 (line 110-115).

(A) Inhibitory effect of IRPS on PRRSV; In the inhibitory assay, at 1.5 h PRRSV infection, cells were treated with different concentrations of IRPS and observe cell status; (B) Preventive effect of IRPS on PRRSV; In the preventive assay, cells were treated with different concentrations of IRPS for 4 h, then 1.5 h PRRSV infection to observe cell status; (C) The direct effect of IRPS on PRRSV; In the direct assay, PRRSV was mixed with various concentrations of IRPS and incubated for 1.5 h to observe cell status.

Point 3: Fig. 6 - need more description

Response 3: We explained and supplemented the relevant contents of Figure 6 (Line 296-301).

“Red represents group I, Blue represents group P, Green represents group M;”.” The red block indicates that the genus was more abundant in that sample than in other samples, and the blue block indicates that the genus was less abundant in that sample than in other samples;”. Comparison box chart of microbial community structure in the genus level. Different asterisks indicate significant differences. * (P<0.05), ** (P<0.01).

Point 4: ne 328 - provide reference

Response 4: We have provided relevant reference literature (line 336-337).

Point 5: Figure 7 - needs explanation in text

Response 5: We explained and supplemented the relevant contents of Figure 7 (Line 367-372).“PRRSV becomes double-stranded RNA (dsRNA) by replication and transcription, acts directly on OAS gene. OAS gene could promote RNaseL expression. RNaseL acts indirectly on DHX33 and MAVS, which in turn activate inflammasomes (NLRP3, ASC, ProCASP1)……….”.

Point 6: Lines 351-358 - better fits in previous paragraph before summary.

Response 6: The position of the sentence (Line 351-358) have changed to Line 349-356.

We are very grateful for the comment and suggestion from reviewer.